# Simple and Robust Detection of *CYP2D6* Gene Deletions and Duplications Using *CYP2D8P* as Reference

**DOI:** 10.3390/ph15020166

**Published:** 2022-01-28

**Authors:** Jens Borggaard Larsen, Steffen Jørgensen

**Affiliations:** 1Laboratory Unit, The Danish Epilepsy Centre, Filadelfia, Kolonivej 11, DK-4293 Dianalund, Denmark; 2Centre for Engineering and Science, University College Absalon, Parkvej 190, DK-4700 Næstved, Denmark; stjo@pha.dk

**Keywords:** *CYP2D6*, *CYP2D7P*, *CYP2D8P*, copy number variation, CNV, genotyping, 5’nuclease assay, HRM, high resolution melting, drug metabolization, pharmacogenetics

## Abstract

Genotyping of the *CYP2D6* gene is the most commonly applied pharmacogenetic test globally. Significant economic interests have led to the development of a plurality of assays, available for almost any genotyping platform or DNA detection chemistry. Of all the genetic variants, copy number variations are particular difficult to detect by polymerase chain reaction. Here, we present two simple novel approaches for the identification of samples carrying either deletions or duplications of the *CYP2D6* gene; by relative quantification using a singleplex 5′nuclease real-time PCR assay, and by high-resolution melting of PCR products. These methods make use of universal primers, targeting both the *CYP2D6* and the reference gene *CYP2D8P*, which is necessary for the analysis. The assays were validated against a reference method using a large set of samples. The singleplex nature of the 5′nuclease real-time PCR ensures that the primers anneal with equal affinity to both the sequence of the *CYP2D6* and the reference gene. This facilitates robust identification of gene deletions and duplications based on the cycle threshold value. In contrast, the high-resolution melting assay is an end-point PCR, where the identification relies on variations between the amount of product generated from each of the two genes.

## 1. Introduction

The cytochrome p450 monooxidases are a superfamily of enzymes, primarily transcribed in the liver. While many of these enzymes have an essential function in biosynthesis and metabolization of endogenous compounds, others are part of the phase I modification/defense system against xenobiotics [1,2]. This last group constitute pharmacological important enzymes, capable of influencing the treatment of patients prescribed substrate drugs [3]. Alterations in their activity can lead to changes in the metabolic pattern, diminishing the effect of the treatment and/or causing adverse drug reactions [4]. Evolutionary, this group of cytochrome P450 enzymes are believed to have arisen to protect their host against environmental and food produced toxins [5]. As a result, these genes are less stable, allowing faster adaptation, and variants in the forms of deletions or duplications occur at a higher frequency than for other members of the superfamily [4].

One cytochrome P450 gene where copy number variations (CNV) are of particular interest is *CYP2D6*. In humans, the subfamily 2D consists of *CYP2D6* and the two pseudogenes *CYP2D7P* and *CYP2D8P*, located together in a 45 kb region on chromosome 22 [6,7]. Although the evolutionary history of this subfamily dates back to before the divergence between amniotes and amphibians, more than 340 million years ago, its present day organization in hominids is more recent [8,9]. Phylogenetic analysis dates the duplication event and separation between *CYP2D6* and *CYP2D8P* back to the divergence between the new world monkeys and Catarrhini, some 18–5 million years ago [7,9]. In contrast, the *CYP2D7P* gene has been acquired more recently by a duplication of *CYP2D6* in the lineage of the great apes. *CYP2D8P* is pseudorized in humans, chimpanzee, and orangutan, by a number of different mutations. In *CYP2D7P,* this is caused by a single frameshift mutation, present in humans and orangutan, but not chimpanzee [7].

The polymorphic nature of the *CYP2D6* gene and its impact on the metabolization of certain drugs had been recognized even before the enzyme was isolated and its gene sequenced [10,11]. Although the enzyme only makes up an estimated 2% of the total content of cytochrome P450 mono-oxidase protein in the liver, it participates in the metabolization of more than 20% of all prescribed drugs [12]. In Caucasians, an estimated 30% of the population has variations in the *CYP2D6* gene, causing changes in metabolic activity [13]. Genotyping of the *CYP2D6* gene is, therefore, a useful tool for predicting the metabolic phenotype and personalizing the treatment of prescribed drugs metabolized by this enzyme. Based on test results, patients are divided into four different groups: poor, intermediate, extensive, and ultrafast metabolizers [10,14]. Ultrafast metabolizers are carriers of one or more fully functional duplication of the gene, while poor metabolizers have two nonfunctional alleles. Abolishment of the enzyme function in an allele can be caused either by single point mutations, or complete or partial deletions of the gene.

Most genotyping methods for copy number variation detection, rely partly or in whole on amplification of the target by PCR [15,16,17,18,19,20,21,22]. Modern real-time PCR techniques are extremely versatile. However, while sensitivity is one of the biggest strengths of this method, it also holds its greatest weakness. Bias can be introduced during amplification that is not only caused by the specificity/annealing temperature of the primers or probes to the target, but may arise due to variations in primer/probe concentrations, the amount of template DNA, or its purity, as well as the concentration of salts or presence of contaminants [23]. This is especially true for assays relying on the multiplexing of primers and probes, or assays that use standard curves or reference samples for relative quantification [24]. 

The most frequently applied method for routine testing for CNV in the *CYP2D6* gene is by relative quantification, using real-time PCR [15,17,19,25]. By this method, primers amplifying a target sequence from the gene of interest are multiplexed with another primer pair targeting a reference gene. The reference gene is commonly a household gene such as *ALB*, *GAPDH*, or *RNase P*, known not to harbor CNV [16,17,19,26]. During amplification, the products of each primer pair are detected in real-time by monitoring the fluorescence generated from different fluorophores. The relative difference between the two cycle thresholds is then compared to that of a reference wildtype reaction, allowing estimation of copy numbers in the sample.

Although relative quantification by real-time PCR is fast and does not demand additional instrumentation, the use of multiplex PCR makes it less robust. Large fluctuations in the amplification efficiency between identical reactions are common, and routine use necessitates multiple duplicates of each sample, in order to obtain a robust result [23,24,27,28]. One way of optimizing these assays is to design multiplex primer pairs with equal amplification efficiency and with a low level of primer dimer formation. Here, we present a novel approach to this problem, using a singleplex real-time PCR reaction for detection of gene deletions and duplications by relative quantification. Furthermore, by using a single universal primer targeting *CYP2D6* and the reference gene *2D8P*, we developed a high-resolution melting (HRM) end-point PCR method that detects these genotypes based on a melt curve analysis. Both methods allow simple and robust detection of deletions and duplications in the *CYP2D6* gene, and thus are of high clinical relevance in relation to pharmacogenetic testing.

## 2. Materials and Methods

### 2.1. Sequence Analysis

The sequences of *CYP2D6*, *CYP2D7P,* and *CYP2D8P* from the latest human genome assembly GHCh38.p13 were downloaded from the National Center for Biotechnology Information (www.ncbi.nlm.nih.gov accession date 11 November 2020) and aligned using Clustal Omega (www.ebi.ac.uk, 11 November 2020). The webtool boxshade (www.expasy.org—discontinued, 11 November 2020) was used to identify similarities between the three sequences in the alignment. Regions with homology between *CYP2D6* and *CYP2D8P*, and containing heterologous bases to *CYP2D7P*, were then identified by visual inspection of the alignment (Table 1).

### 2.2. Design of Primers and Probes for the 5′Nuclease Singleplex PCR

Based on the generated alignment, a universal primer set for *CYP2D6* and *CYP2D8P*, but not *CYP2D7P*, was designed targeting exon 9. A general set of rules for the primer design was used, which included a theoretical annealing temperature of 57–63 °C, primer size of 18–25 nt, and 3’ destabilization, by minimizing the number of C’ and G’ in the last five bases. Bases heterologous to the sequence of *CYP2D7P* were placed in the 3’ end of the primers, if possible, and the amplicon size was minimized. 

Two gene specific 5′nuclease probes were designed; one targeting the amplified sequence of *CYP2D6,* and one the sequence of *CYP2D8P*. The probes were designed to have an annealing temperature as close to each other as possible, and a theoretical annealing temperature 6–10 °C higher than that of the primers. A guanine base immediately adjacent to the 5’ labelling was avoided. 

Before ordering, each primer and probe sequence was validated in silico using the online software tool Netprimer (Premier Biosoft, http://www.premierbiosoft.com/netprimer/, 11 November 2020), and their specificity to their intended targets was confirmed by performing a Primer-Blast search against the GHCh38.p13 assembly (National Center of Biotechnology Information —www.ncbi.nlm.nih.gov/tools/primer-blast/, 11 November 2020). *CYP2D6* probes were labelled 5′ with FAM, while that of *CYP2D8P* was labelled with CalFluor 560. For quenching of the fluorochrome signal in the intact probe, the 3′ end was labelled with black hole quencher-1 (BHQ-1). All primers and probes were ordered from LGC-Biosearch (Aarhus, Denmark). 

### 2.3. Design of Primers for Use in the High Resolution Melting PCR Assay

To facilitate detection of deletions and duplications using melt curve analysis, a single universal primer was developed capable of amplifying from both the sequence of *CYP2D6* and *CYP2D8P*. As complementary primers, a gene-specific primer was designed for each of the two intended targets. These gene-specific primers generate amplicons variating in size and melt temperature, thereby facilitating separation during analysis. To predict the difference in melt temperatures during the design of the primers, the online tool ‘Oligo Calc’ (http://biotools.nubic.northwestern.edu, 11 November 2020) was used. The design followed the same rules as for the relative quantification, and the primers were validated *in silico* accordingly.

### 2.4. Reference Samples and DNA Extraction

Material submitted to the Laboratory of the Danish Epilepsy Centre and tested using a previously published method was used as reference for validation of the assays. Patient samples were anonymized upon retrieval and DNA extraction was performed using a MagNA Pure Compact Nucleic Acid Isolation Kit I (Roche Diagnostic, Basel, Switzerland) applying 200 µL of full blood and a 100 µL elution volume. Following the extraction, samples were quantified using a NanoDrop One spectrophotometer (Thermo Fischer, Waltham, MA, USA), and the DNA concentration was normalized to 20 ng/µL.

### 2.5. Assay Validation

In total, 48 samples were selected for the validation, based on the genotyping results obtained by the reference method. These samples consisted of 18 tested as wildtype, 20 identified as carriers of a *CYP2D6* deletion variant (*5), and 10 containing one or more duplications of the *CYP2D6* gene.

The applied routine method for CNV genotyping of the reference samples (in the following referred to as *CYP2D6:RNase P*) is accredited by the Danish Accreditation Fund (DANAK) according to the ISO 15189:2013 standard. It relies on relative quantification real-time PCR by amplification of a target sequence in exon 9 of the *CYP2D6* gene. The method is modified from Schaeffler et al. (2003), in that it is multiplexed with a commercial CNV reference kit targeting the *RNase P* gene, instead of the albumin gene described in the original publication [17]. For this study, the assay was performed on duplicates of each sample, in a 20 µL reaction volume containing 10 µL Bio-Rad ITaq Universal Probes Supermix (Bio-Rad Laboratories Inc., Hercules, CA, USA—cat# 1725131), 250 nM of each of the primers of the CYP2D6 assay [17], 150 nM of the probe, 1× RNase P primer/probe mix (Thermo Fischer, Waltham, MA., USA cat# 4403326), and 4 µL of template DNA (20 ng/µL). Temperature cycling was performed in a Bio-Rad CFX Connect real-time PCR instrument, running a program consisting of 3 min at 95 °C, followed by 35 cycles of sequential denaturation for 15 s at 95 °C, and 1 min of elongation at 60 °C. Results were calculated using the 2^−∆∆Ct^ method in the CFX manger v3.1 software (Bio-Rad Laboratories Inc.), by averaging the values from a set of six wildtype samples, used as calibrators and included in each run.

### 2.6. Detection of CYP2D6 Deletions and Duplications by the 5′Nuclease Singleplex PCR Assay

The singleplex assay *CYP2D6:2D8P* was performed in 20 µL reaction volumes, containing 10 µL of 2× ITaq Universal Probes Supermix, 250 nM of each of the two primers, 200 nM of each of the probes, together with 4 µL of Template DNA (20 ng/µL). Temperature cycling was performed, applying the same instrument, program, and software setup as for the reference multiplex PCR assay described above. The only exception was that four calibrators instead of six were used per setup.

The PCR efficiency of the *CYP2D6:2D8P*, and the *CYP2D6:RNase P* assay used as reference, was evaluated using standard curve regression. A ten-time dilution series of a wildtype sample, going from 4 ng/µL to 0.004 ng/µL DNA, was generated, and PCR was performed in duplicates using the same reaction concentrations and cycle conditions as above. Using the software CFX Manager v3.1, the PCR efficiency of both probes in each of the two assays was calculated by plotting the logarithmic concentration of the added template DNA against the cycle threshold Cq.

### 2.7. Detection of CYP2D6 Deletions and Duplications by High-Resolution Melting

For detection of deletions and duplications using HRM, the concentration of the universal forward primer was first optimized against a standard concentration of 250 nM for each of the gene specific primers. The optimization established a concentration at which both of the two amplicons, deferring in size, gave a comparable intensity during melt curve analysis. The optimal concentration of the universal primer was 200 nM for the assay. PCR was performed in 20 µL reaction volumes with 10 µL of Precision Melt Supermix (Bio-Rad cat#1725110) and 2 µL of Template DNA (20 ng/µL), using a Bio-Rad CFX Connect instrument. PCR amplification consisted of an initial denaturation at 95 °C for 3 min, followed by 40 cycles of denaturation at 95 °C for 15 s, annealing of the primers at 60 °C for 30 s, and elongation for 30 s at 72 °C. A melt curve analysis was then performed by first incubating the samples for 3 min at 60 °C, and measuring fluorescence at 0.2 °C/s steps from 70–95 °C. The software package Precision Melt Analysis v1.2 (Bio-Rad Laboratories) was used for normalization of the fluorescence peaks between the samples, and for grouping of the samples according to their melt curve pattern.

### 2.8. CYP2D6 and CYP2D8P Sequencing

The target region of the singleplex and the HRM assays, were selectively amplified from the *CYP2D6* and *CYP2D8P* gene, using the gene specific primers listed in Table 2, C. PCR was performed in 25 µL reaction volumes with 12.5 µL of CloneAmp HiFi PCR premix (TakaraBio, cat#639298), 250 nM of each primer, and 100 ng genomic DNA. PCR amplification was performed with a Bio-Rad CFX96, using the following conditions: 35 cycles of 98 °C for 10 s, 60 °C for 10 s, and 72 °C for 30 s. The PCR products were gel purified using a 1 % agarose gel. Bands, corresponding to 676 bp for the *CYP2D6* fragment and 670 bp for the *CYP2D8P,* were sequenced by Eurofins Genomics, using the same gene-specific primers as for the PCR amplification.

## 3. Results

### 3.1. Design and Validation of the 5′Nuclease Singleplex PCR Assay

Sequence alignment identified a common region suitable for designing universal primers, facilitation specific amplification from the *CYP2D6* and *CYP2D8P* genes, but not from *CYP2D7P*. The targets of the primers are located in exon 9, starting at position 42126676 of the *CYP2D6* sequence and position 42149988 of *2D8P* (Table 1). This region is the same as that covered by the assay designed by Schaeffler et al. (2003). The primer pair flank a sequence with heterogeneity between the *CYP2D6* and *CYP2D8P*, thereby facilitating design of specific probes against each of the two genes (Table 2).

The equal affinity of the primers for both gene targets, and the efficiency of the PCR assay, was investigated by comparing a standard curve to that from the reference assay (Figure 1). The obtained slope values for the *CYP2D6:2D8P* assay where −3.39 for the *CYP2D6* probe and −3.37 for the *CYP2D8P*, resulting in a calculated PCR efficiency of 97.3% and 97.9% (Figure 1A). The same analysis for the reference *CYP2D6:RNase P* assay gave a slope of −3.18 and −3.40, respectively, and a calculated efficiency of 105.4% and 96.9% (Figure 1B).

The *CYP2D6:2D8P* assay was validated against a large set of reference samples (*n* = 48). As a method for comparing the result from the *CYP2D6:2D8P* to that of the reference, the relative quantification (RQ) values obtained by each of the assays were plotted against each other (Figure 2A). This plot clearly separated the samples into the three genotypes (deletion *5, wildtype, duplication *xN*) and showed 100% consistent results between the two assays. The two assays where further compared by calculating the mean RQ for each of the three genotype groups (Figure 2B). The RQ mean for the reference samples analyzed by the *CYP2D6:2D8P* assay was 0.45 for the deletion *5 allele, 0.94 for the wildtype, and 1.90 for the duplication. In comparison, the values for the *CYP2D6:RNase P* were 0.41, 0.80, and 1.43 for the same set of samples.

As both *CYP2D6* and *CYP2D8P* are competing for the same set of primers in the assay, the separation between the genotypes would be expected to increase throughout the PCR reaction. This was verified at the end-point of the PCR for the *CYP2D6:2D8P* assay, by plotting the relative fluorescence from the probes against each other (Figure 3).

### 3.2. Design and Validation of the High Resolution Melting Assay

The developed genotyping technique using high-resolution melting, relies on a single primer targeting a sequence conserved between the target gene *CYP2D6*, and the reference gene *CYP2D8P*. The single universal primer is multiplexed in the PCR reaction with two gene specific primers, designed to generate amplicons differing in length/melt temperature of the double stranded DNA. This allows separation of the peaks generated by the *CYP2D6* and *CYP2D8P* gene on the melt curve plot (Figure 4). As the universal primer is added at a lower concentration, making it the rate-limiting factor of the PCR, any difference in the starting template amounts between the two gene targets is expected to increase throughout the PCR amplification. Detection of deletions and duplications was thus performed by comparing the melt curves to controls, and looking at the differences in fluorescence, caused by the proportional shift between the amplicons of *CYP2D6* and *2D8P* (Figure 4A).

The proposed method was tested by designing an assay targeting the same region in *CYP2D6* exon 9 and applying the same universal forward primer as for the assay developed for relative quantification in this study (Table 1). Gene specific primers were designed located immediately downstream, resulting in a fragment size of 60 bp for the *CYP2D6* gene, and 68 bp for *CYP2D8P* (Table 2). Optimization was performed on a wildtype sample, to ensure comparable peak heights for both amplicons in the melt curve analysis for this genotype (Figure 4A). This was done by lowering the concentration of the universal primer alone. When tested on samples carrying a deletion or duplications, this resulted in a change in the melt curve pattern, allowing easy discrimination between the three genotypes (Figure 4).

The assay was further validated on the same set of reference samples as used for the relative quantification. The reference samples carrying deletions *5, and homozygote wildtypes, gave a result 100% consistent with the previous findings (Figure 4B). Four of the duplicate samples showed a slightly deviating melt curve pattern. In order to identify the reasons for these discrepancies, this region from both the *CYP2D6* and the *CYP2D8P* gene was sequenced. One of the samples carried a C > T transition within the assay region (Table 1B—*CYP2D8P*—location 42150006), possibly explaining the different melt curve pattern. The discrepancy for the three other samples could not be resolved.

## 4. Discussion

The purpose of the present study was to develop a simple and robust assay for routine pharmacogenetic detection of copy number variations in the *CYP2D6* gene. Previously, the use of *CYP2D8P* as a reference gene for CNV detection of *CYP2D6* was reported in two other studies. While Söderbäck et al. first described a method for genotyping *CYP2D6* by pyrosequencing, Nakamura et al. developed a method combining loop mediated isothermal amplification (LAMP-PCR) and electrochemical detection [21,29]. Both of these studies applied proprietary techniques that require additional instrumentation as well as reagents. In contrast, the methods reported in the current study rely on simple real-time PCR technology. The novelty of this, therefore, relies on two factors:The equal affinity and efficiency of the universal primers when amplifying the two targets located in the *CYP2D6* and *2D8P* gene.The competition of the two targets for the same pool of universal primers, in the later phase of the PCR amplification.

The first is important when performing relative quantification using real-time PCR, as the method relies on cycle threshold values (Cq) obtained during the exponential phase of the reaction. During this phase, the primers and other reagents are not the rate-limiting factor, but rather the efficiency of the primers and the concentration of template DNA. By reducing the number of primers and by having primers with a 100% similar annealing temperature to the two targets, an equal amplification efficiency is obtained for *CYP2D6* and *CYP2D8P* (Figure 1).

The second benefit is a prerequisite for developing an end-point PCR method for identifying deletions and duplications. While a regular multiplex PCR has two (or more) different pools of primers and the depletion of one pool will tend to have no direct effect on the amplification performed by the other, this is not the case using universal primers. Here, following the exponential phase, the two targets will begin to compete for the same pool of free universal primers in the PCR reaction. As this pool becomes depleted, the most abundant template will have a continually increased chance of a primer annealing and initiating amplification. When applied to a sample carrying deletions or duplications in the *CYP2D6* gene, this results in a continuously greater difference between the signal of the target and the reference gene. This principle is applied in the HRM assay, where the identification is based on the difference in fluorescence obtained during melting of the two amplicons (Figure 4). However, it may also be used to develop similar end-point PCR assays based on other chemistries, such as 5′nuclease/Taqman (Figure 3). This has the potential to reduce the overall cost of PCR-based genotyping of the *CYP2D6* gene, by allowing the same workflow and setup for identification of both gene deletions/duplications and single nucleotide polymorphisme (SNP) [30,31]. However, in order for this to work in a 5’nuclease assay, an additional separation between the signal strength of the two probes is required. This is particularly true in order to be able to differentiate between the wildtype and samples harboring duplications of the *CYP2D6* gene (Figure 3). One possible solution could be to use double quenched probes.

Since the clinical relevance of exact copy number calls of duplications is very limited, both assays in their present form are qualitative. This is especially true for the HRM assay, as it is an end-point PCR. In contrast the 5′nuclease singleplex assay relies on the real-time Cq values obtained during the exponential phase of the PCR. With further development, the method, therefore, ought to be useful for exact quantification of duplicates in the *CYP2D6* gene, should that be warranted. This would, however, necessitate a more stringent and elaborate setup than is generally feasible for routine pharmacogenetic testing.

A complicating factor when genotyping the *CYP2D6* gene is the possible presence of partial deletions/duplications or hybrids between the *CYP2D6* and the *2D7P* gene. The location of the two assays presented here allows the discrimination of one common hybrid allele *36 found in tandem CNV. This is a non-functional hybrid allele containing a conversion to *CYP2D7P* within or upstream of exon 9. Therefore, if the target sequence of the assay is upstream from the conversion site, it will be genotyped as a wildtype. In contrast, an assay targeting a region downstream would identify it as a deletion and thereby, correctly, as a non-functional allele [16].

Although the design of an assay applying two universal primers adds restrictions, by lowering the number of possible target sequences in the *CYP2D6* gene, the initial sequence analysis identified other possible locations. One is around the same area in exon 6 targeted by Söderbäck et al. [29]. Other regions of the *CYP2D6* gene may, however, be made accessible by incorporation of single degenerate or modified bases, such as inosine or locked nucleic acids, into the primer sequence.

In contrast to the 5′nuclease singleplex assay, the HRM method only relies on a single universal primer. This increases the potential targets for such assays in the *CYP2D6* gene significantly. The method may, therefore, also be used for detecting deletions and duplications in other genes i.e., *SULT1A1* or *UGT2B17* [28,32].

In conclusion, the novel methods presented in this study allow fast and robust identification of deletions and duplications in the *CYP2D6* gene, applying standard real-time PCR detection techniques. The outlined approach may also be applicable to other PCR chemistries used for genotyping. In addition, if suitable reference sequences can be identified, these methods could help simplify the routine identification of deletions or duplications in genes other than *CYP2D6*.

## Figures and Tables

**Figure 1 pharmaceuticals-15-00166-f001:**
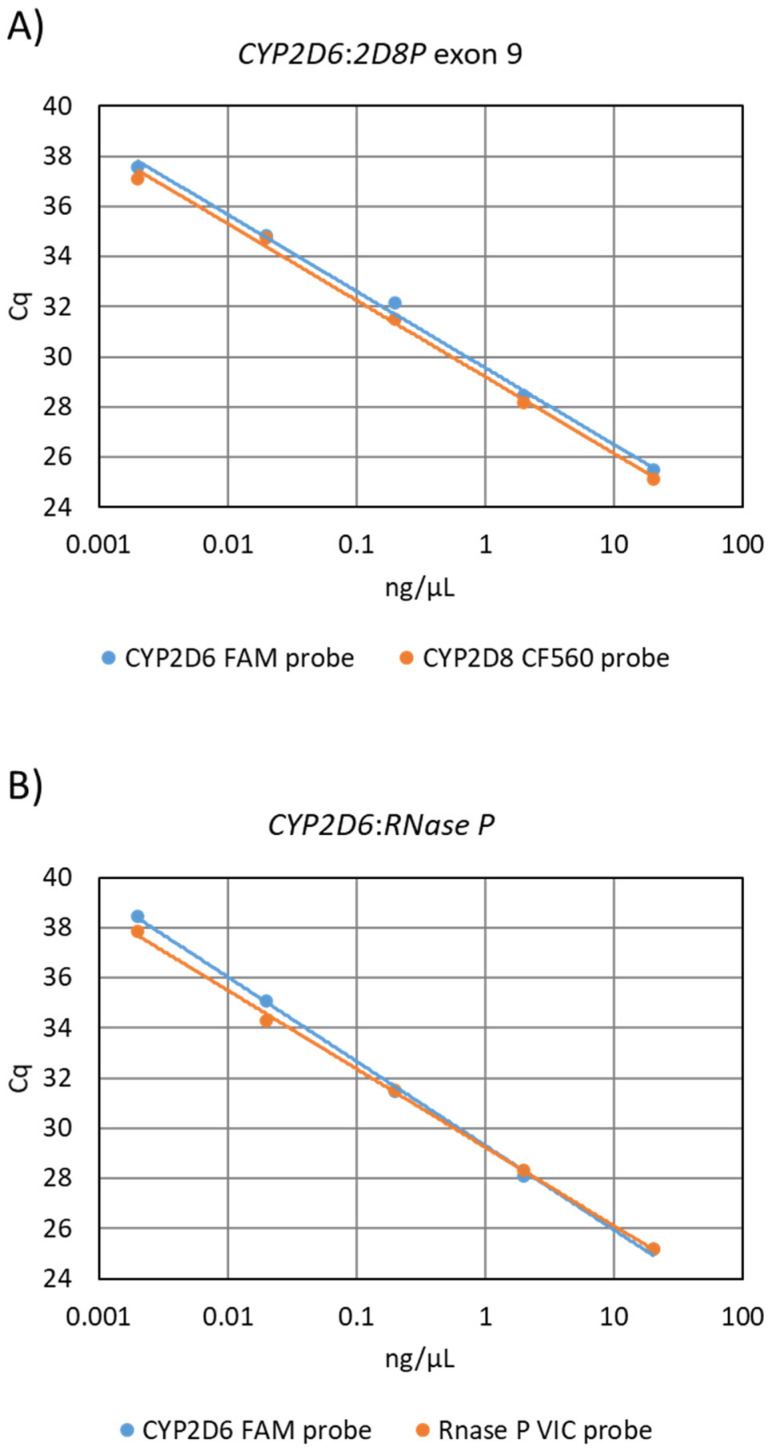
Analysis of primer amplification efficiency by use of the standard curve method, performed by dilution of a wildtype samples. (**A**) Standard curve generated by the 5′nuclease sing-leplex *CYP2D6:2D8P* assay. Slope of the *CYP2D6* probe −3.39 efficiency 97.3%, *CYP2D8P* probe slope −3.37 efficiency 97.9% (**B**) Standard curve generated using the CYP2D6:RNase P reference assay. CYP2D6 probe slope −3.18 and 105.4% efficiency, RNase P probe slope −3.40 calculated efficiency 96.9%.

**Figure 2 pharmaceuticals-15-00166-f002:**
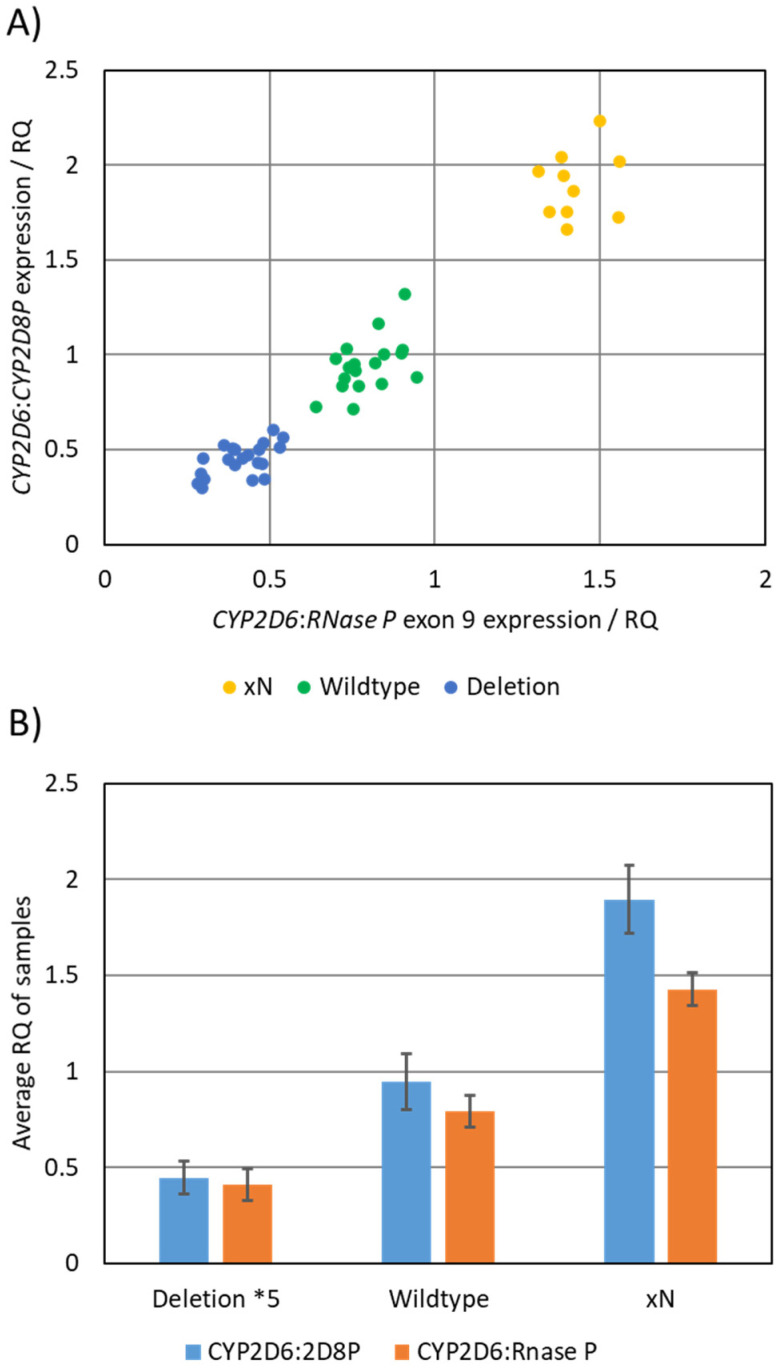
Comparison between the *CYP2D6:2D8P* singleplex, and the *CYP2D6:RNase P* reference assay. (**A**) Graphical representation of results obtained by parallel testing, where the calculated RQ values obtained from the two assays are plotted against each other. The plot separated the samples into three groups based on the copy number of the sample. (**B**) Calculated average RQ value for the samples carrying deletion *5 (*n* = 20), wildtype (*n* = 18), and duplications (*n* = 10).

**Figure 3 pharmaceuticals-15-00166-f003:**
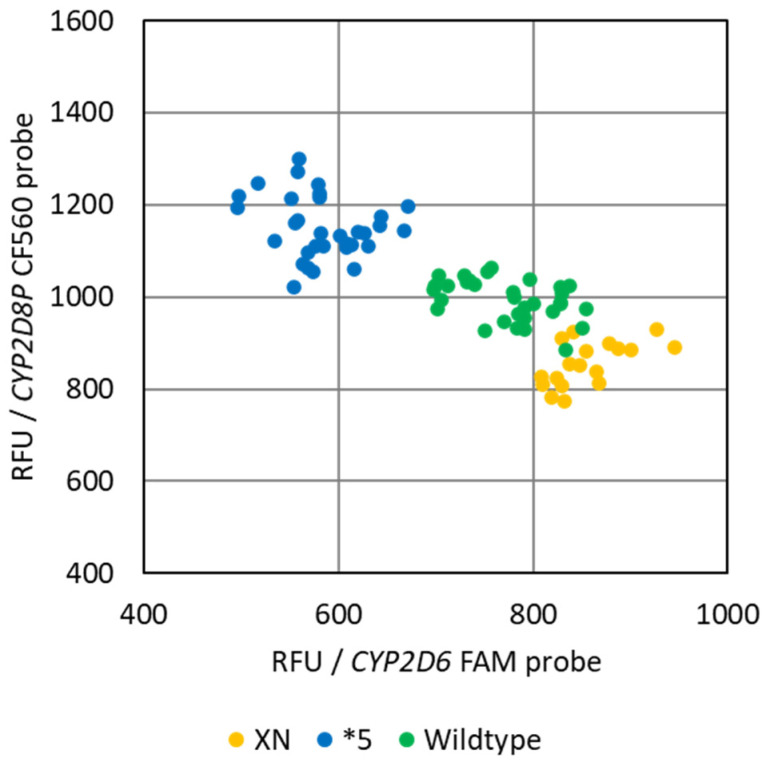
Common end-point PCR allele discrimination plot of the reference samples obtained by the singleplex *CYP2D6:2D8P* assay. The plot shows the differentiation into the three genotype groups, deletions *5, wildtype, and samples with one or more duplicates of the CYP2D6 gene (xN).

**Figure 4 pharmaceuticals-15-00166-f004:**
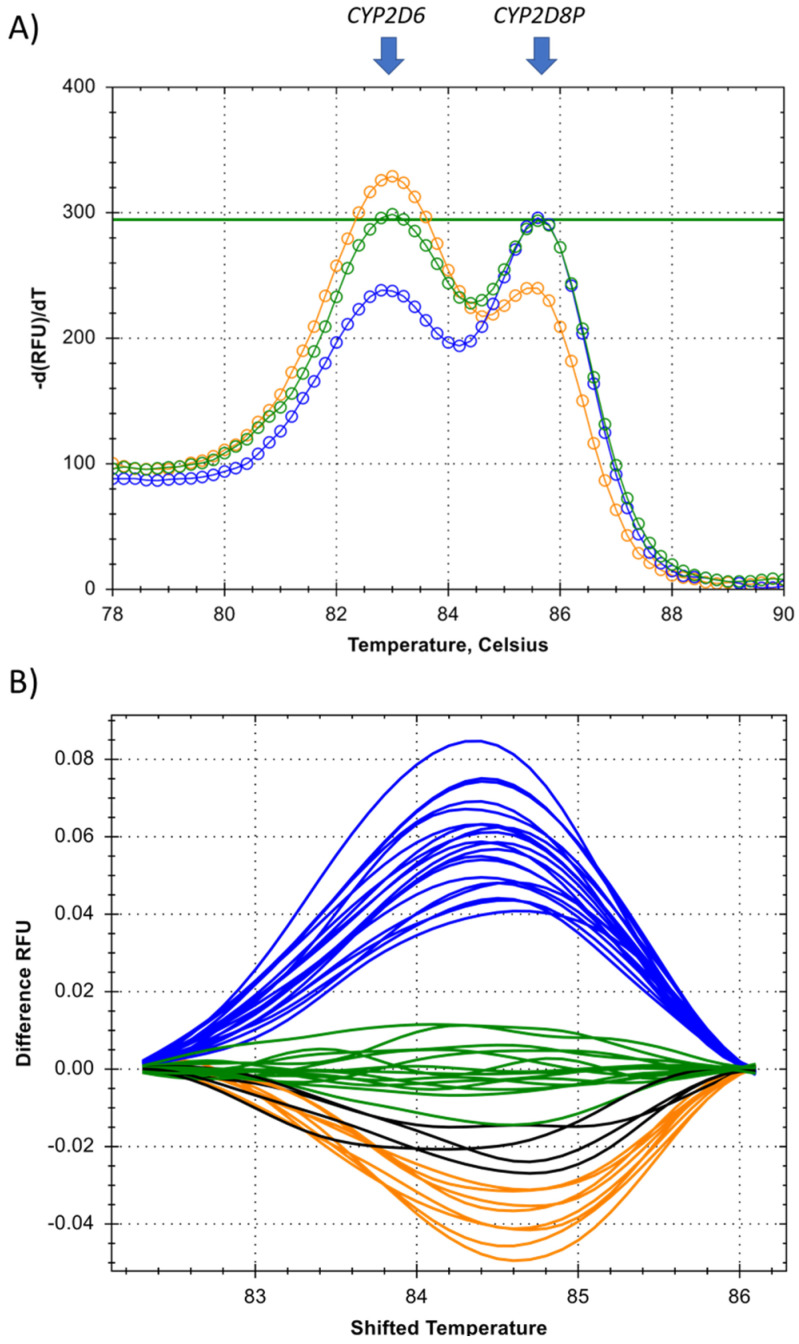
Detection of *CYP2D6* deletions or duplications by high-resolution melting PCR. (**A**) Melt curve showing the proportional change between the two amplicons generated from the *CYP2D6* and the *CYP2D8P* gene, identified by the two arrows. (**B**) Temperature shifted view of the melt curves from all of the reference samples (*n* = 48) used for the validation of the assay. Clustering was done by automatic calling, using the Precision Melt software from Bio-Rad. Blue = deletions *5, gree*n* = wildtype, orange = duplications, and black = unresolved duplicate samples.

**Table 1 pharmaceuticals-15-00166-t001:**
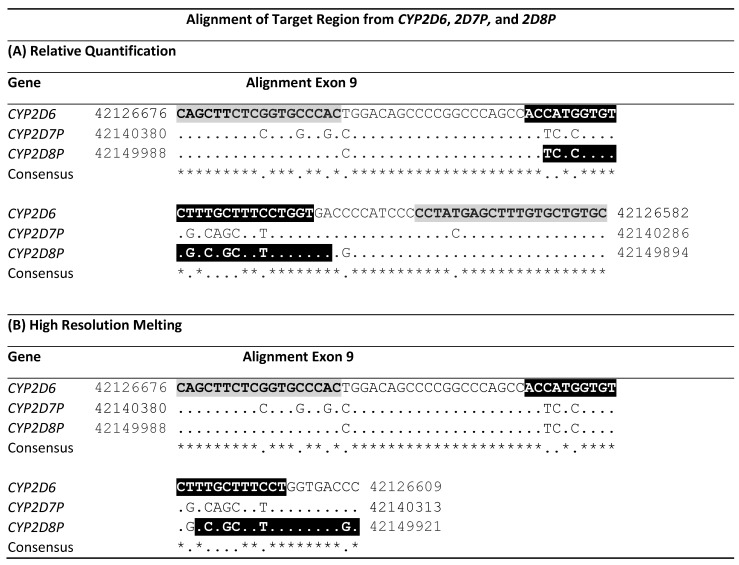
Alignment of the primed region in exon 9 from *CYP2D6* and the two pseudogenes, *2D7P* and *2D8P*. The alignment is oriented 5′-3′ to the gene. Start and end position of the sequences in the GRCh38.p13 assembly are shown. Letters with gray background mark the location of the *CYP2D6:2D8* universal primers, while the locations of the two gene specific probes/primers are shown in black reversed letters. ‘*’ Indicates conserved bases in the alignment. (A) shows the location of the primers and probes for the 5′nuclease singleplex assay. (B) shows the location of the primers designed for high-resolution melting. Background color shows the location of primers and probes.

**Table 2 pharmaceuticals-15-00166-t002:** Primers and probes designed and used in this study. ‘Size’ is the size of the amplicons produced by the primers. (A) Primers and probes designed for the 5′nuclease singleplex *CYP2D6:2D8P* PCR assay for use with relative quantification. Probe labelling: FAM = Fluorescein, CF560 = Calfluor 560, BHQ-1 = Black Hole Quencher 1. (B) Primers designed for detection by high-resolution melting. (C) Gene specific primers designed for sequencing of the *CYP2D6* and *CYP2D8P* target regions of the singleplex and high-resolution melting analysis.

**Primers and Probes Designed for this Study**		
**(A) Primers and probes for relative quantification**		
**Name**	**Sequence**	**Size**
CYP2D6:2D8 exon 9 Fwd	5′ CAGCTTCTCGGTGCCCAC 3′	95 bp
CYP2D6:2D8 exon 9 Rev	5′ GCACAGCACAAAGCTCATAGG 3′	
CYP2D6 exon 9 probe	5′ FAM-ACCAGGAAAGCAAAGACACCATGGT-BHQ1 3′	
CYP2D8 exon 9 probe	5′ CF560-TCACCAGAAAGCCGACGACACGAGA-BHQ1 3′	
**(B) Primers for high-resolution melting**		
**Name**	**Sequence**	**Size**
CYP2D6:2D8 exon 9 Fwd	5′ CAGCTTCTCGGTGCCCAC 3′	
CYP2D6 exon 9 Rev	5′ AGGAAAGCAAAGACACCATGGT 3′	60 bp
CYP2D8 exon 9 Rev	5′ GCGTCACCAGAAAGCCGA 3′	68 bp
**(C) Primers for sequencing**		
**Name**	**Sequence**	**Size**
CYP2D6 Seq Fwd	5′ GTCTAGTGGGGAGACAAACCA 3′	676 bp
CYP2D8 Seq Fwd	5′ CTAGTGGGGAAGGCAGACCA 3′	670 bp
CYP2D6:2D8 Rev	5′ GCACAGCACAAAGCTCATAGG 3′	

## Data Availability

Data is contained within this article.

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
