# Peer review of "Simple and Robust Detection of CYP2D6 Gene Deletions and Duplications Using CYP2D8P as Reference"

_pharmaceuticals, 2022, doi:10.3390/ph15020166_

Round 1

Reviewer 1 Report

In this manuscript Larsen et al. described accurately the novel approache for CNV detection of the CYP2D6 gene using CYP2D8P as reference gene by a singleplex 5’nuclease real-time PCR assay. The authors fully detailed the protocol and the validation process used to perform this new fast approach for CNV detection through high resolution melting allowing the opportunity to make available a cost efficient test for CNV detection of the CYP2D6 gene. This approach achieves low variability in CNV detection and might be applied to other genotyping approaches. The manuscript is acceptable in this form for publication after a fine revision of English language and style. 

Author Response

Reply to the Reviewer

We are very happy and thankful for the comments. We have corrected the manuscript accordingly, and refined the language.

The Authors

Reviewer 2 Report

The authors present in this manuscript an interesting procedure(s) enabling simple CYP2D6 CNV analysis.

The manuscript has merits that mainly rely on the use of a CYP2D8 as a reference for normalisation and on a single primer pair. It is interesting to note that because the two targets are in competition for primer annealing a  larger difference in PCR signals is thought to be achieved. However, this may result in bias on the real estimation of the CNV (overestimation???).   Therefore the authors need to show the performance of this system non only for deletion and single duplication events but also in the occurrence of defined  multiduplication events. For this reason, the materials and methods section need to report the features(single duplication, multiduplication or deletion) of DNA samples used for validation. 

Author Response

Reply to the Reviewer

We are grateful for the comments. However, we disagree on the need for the suggested additional experiment for the following reasons.

In the paper, we describe two benefits of the invention.

1.       Relates to the equal efficiency of the primers for each of the two targets (CYP2D6 and 2D8).

2.       The competition of the two targets for the same pool of primers, as the PCR reaction progress.

The first is important when performing relative quantification using real-time PCR. This method relies on Ct values obtained during the exponential phase of the reaction. In this phase, the primers or the other reagents are not the rate-limiting factor, but rather the efficiency of the primers and the concentration of template DNA. By reducing the number of primers and by having primers with a 100% similar annealing temperature to the two targets, we obtain the same amplification efficiency for both of them (Figure 1). Therefore, this method should be very well suited for correct call of duplication number should that be needed.

The second benefit of the invention is the increased separation in the signal from samples that carries either a deletion, wildtype, or duplication, as they compete for the same pool of primers (figure 3 and 4). This principel is necessary for the development of the HRM end-point PCR, where the one common primer becomes the rate-limiting factor. The competition for this primer ensures that any difference in the initial template concentrations between the targets are enlarged. As we suggest in the discussion and show in figure 4, it may also be used for developing an allelic discrimination assay for detection of CNV based on a singleplex 5’nuclease assay (as when using commercial Taqman assays for SNP genotyping). 

While an end-point PCR is a fast and very convenient method for routine detection of CNV, it is primarily a qualitative assay. We therefore agree with the reviewer that this method does not allow precise call of the copy number from samples with multiple duplications, without further development of the assay. 

However, no matter which method is used (including real-time PCR assays not described in this manuscript), correct call of duplication numbers demands a very stringent setup to be accurate, including complete calibration curves, as well as multiple replicates of each sample. It is therefore not well suited for large volume routine testing.  Furthermore, although exact quantification of the number of duplications may hold some academic interest when reporting the genotype, its clinical relevance in relation to the phenotype from the pharmacogenetic test is small. To our knowledge no lab (at least within Scandinavia), report exact copy number from CYP2D6 pharmacogenetic tests.

Therefore, although the suggested additional experiment may hold some advantages when further developing or commercializing the assays, we disagree with the reviewer regarding its importance for the present study, as it does not add any additional clinical relevance to the test.

The Authors

Reviewer 3 Report

This manuscript describes the development of robust molecular methods to detect CNV of CYP2D6 gene. This is a very important contribution due to the difficulty of this genotyping and the relevance that CNV of CYP2D6 represent to achieve efficacy in clinically prescribed drugs that are substrates of CYP2D6. It is very well written and presented.

Only please correct the classification of CYP 450 enzymes as the two articles cited  at the beginning of the Introduction do not mention the one described in the manuscript. You could use https://doi.org/10.2174/1389450118666170125144557.

Author Response

Reply to the Reviewer

We are very thankful for the comments, and pleased to read that the reviewer agrees with us regarding the importance of this contribution. We have edited the manuscript, rewritten the first paragraph and added additional references.

The Authors

Round 2

Reviewer 2 Report

Dear authors, I understand your position. The clinical aspect is prevalent in such method. However, you are trying to propose this procedure for copy number variation and not for merely duplication detection; additionally, one could hypothesize,erroneously, to use this procedure for CNV detection and not only for pharmaciesnetic assays.  Thus, in my opinion additional experiments should be performed.

Alternatively, you have to modify your text eliminating all the possible misunderstanding including title and main text. Additionally such critic aspect need to be properly discussed along the manuscript.

Author Response

Dear Reviewer

We appreciate your point, and have tried to accommodate it into the revised manuscript. This has been done by changing the title, editing the abstract/end of the introduction, and adding an additional paragraph to the discussion, as well as additional corrections. Given the novelty of our approach and its clinical relevance, we hope that you will find the revised form of the manuscript acceptable, and suitable for publication.

Sincerely

The Authors

Round 3

Reviewer 2 Report

The manuscript now presents the issues that I required. Well done